# SLAD : Shared LoRA Adapters for Task Specific Distillation

## Abstract

In the context of resource-constrained environments such as embedded systems, adapting reduced-size foundation models to downstream tasks has become increasingly popular. This has recently motivated the emerging setting of task-specific distillation, where a larger and a smaller version of the same foundation model are both adapted to the same downstream task, with the goal of transferring knowledge from the former to the latter. Recent work has demonstrated the benefits of using a larger version of the same foundation model to assist the adaptation of a smaller one. Typically, the larger model (teacher) is first adapted via fine-tuning or linear probing before its knowledge is distilled into the smaller model (student). While fine-tuning the teacher often increases its performance, recent work showed that probing it leads to better knowledge distillation to the student. Our findings show that this is mainly due to a mis-alignment in feature representation between the teacher and the student which occurs during the fine-tuning of the teacher. Inspired from existing efforts to preserve previously learned knowledge, we first propose to leverage low-rank adaptation, resulting in better feature alignment and therefore better knowledge transfer. Drawing from this insight, we propose to further enhance the feature alignment, through a parameter-sharing strategy of the adapters between the two encoders during joint-training. Our proposed method SLAD shows a better feature alignment between the teacher and student which results in increased performance for not only the student but also for the teacher model while being $2\times$ faster to train than fine-tuning. Through extensive experiments on multiple datasets of classification and segmentation tasks, we demonstrate the improved accuracy and transfer efficiency of our method achieving state-of-the-art performance in the task-specific distillation framework.[1]

## 1 Introduction

The rise of foundation models trained via self-supervised learning (SSL) has allowed significant advances in deep learning (Oquab et al., 2023; He et al., 2022; Radford et al., 2021; Fang et al., 2024), enabling the creation of versatile models capable of generalizing across a wide range of tasks. These models can often reach state-of-the-art performance when adapted to specific tasks through techniques such as fine-tuning, linear probing, or parameter-efficient fine-tuning (PEFT) (Lialin et al., 2024; Han et al., 2024).

However, in resource-constrained environments, such as embedded or edge devices, the high computational and memory requirements of large foundation models make their deployment impractical (Xu et al., 2024). This motivates the use of smaller foundation models, which are increasingly made available through model compression techniques such as knowledge distillation (Hinton et al., 2015). Due to their limited capacity, these compact models are typically not trained from scratch but rather derived from larger models (Oquab et al., 2023; Zhang et al., 2023; Wu et al., 2022).

Yet, compressing foundation models while preserving their versatility and performance remains a major challenge (Oquab et al., 2023). These models are designed to be general-purpose, encapsulating broad and diverse knowledge learned from their pre-training web-scale datasets. Reducing their

---

[1]Code to be released at: `https://anonymous.4open.science/r/SLAD-Shared-LoRA-Adapters-for-Task-Specific-Distillation-39DC`

size inherently limits their representational capacity, often leading to significant performance drops when adapting to downstream tasks like image classification or segmentation, the focus of this work.

This challenge has recently motivated the emerging setting of task-specific distillation, where a larger and a smaller version of the same foundation model are both adapted to the same downstream task, with the goal of transferring task-relevant knowledge from the teacher to the student. To mitigate this degradation, recent approaches have explored the use of larger foundation models to assist in the adaptation of their smaller counterparts, while discarding the large models at inference time. For instance, methods such as TinyBERT (Jiao et al., 2020) in NLP and G2SD (Huang et al., 2024) in computer vision follow multi-stage pipelines involving fine-tuning the teacher model and subsequently distilling its knowledge into a student model. The teacher and student pair are typically pre-trained using the same routine and of different sizes (Jiao et al., 2020; Marrie et al., 2024).

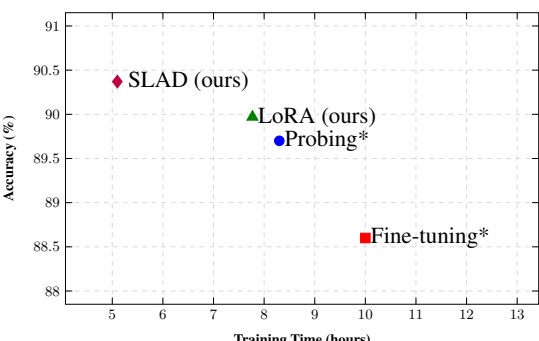

Figure 1: Comparison of accuracy vs. training time for different methods on the CUB Dataset. * refer to methods from Marrie et al. (2024)

Recently, Marrie et al. (2024) showed that while fine tuning increases the performance of the teacher, it does not necessarily benefit the student during the distillation process, showing that a simple linear probing of the teacher results in better knowledge distillation. This indicates that a more expressive model does not automatically translate into a better teacher. However, although linear probing the teacher achieves better knowledge transfer, it remains a shallower model compared to fine-tuning the entire teacher on the task. This limits the teacher's capacity to capture task specific information which is often better captured through fine-tuning (Bensaid et al., 2024; Kerssies et al., 2024). Hence, we ask ourselves the question: **how can we train a more expressive teacher while ensuring better task-specific knowledge distillation for the student?**

To do so, we start by performing an analysis on the feature alignment ((Kornblith et al., 2019)) between the teacher and the student. Our analysis reveals that fine-tuning the teacher results in a significant drop in feature alignment between the two models, indicating a potential harmful mismatch leading to less efficient distillation. This effect is well known in the continual learning literature, where fine-tuning may change the feature representations of the pre-trained model, resulting in catastrophic forgetting (Kirkpatrick et al., 2017).

One common approach to balance the trade-off between catastrophic forgetting and downstream task performance consists in leveraging low-rank adapters (Hu et al., 2022) (LoRA). Inspired by previous efforts in this literature, we thus first propose to use LoRA adapters to improve feature alignment. This improvement in alignment is confirmed by our experiments and leads to improved performance after knowledge distillation, compared to both fine-tuning and linear probing. However, fine-tuning with LoRA does not explicitly enforce the feature alignment between the two models.

In order to further enforce feature alignment, we propose to share the weights of the LoRA adapters between the teacher and the student encoders and to train them jointly. By sharing a subset of the teacher's adapters with the student, we further enforce feature alignment throughout the two encoders. Thanks to this parameter-sharing strategy, SLAD achieves a better feature alignment, resulting in better performance through the knowledge distillation process. Our experiments reveal that our joint training strategy also benefits the teacher, allowing it to reach higher performance than a fully fine-tuned teacher. We thus obtain both a more expressive teacher and a better knowledge distiller. Additionally, by jointly training both encoders, SLAD can be performed in a single step, resulting in faster training times compared to popular alternatives.

Through extensive experiments on classification and segmentation tasks, we show the effectiveness of the proposed SLAD method, which achieves consistent gains over the existing state-of-the-art for task-specific distillation, with an absolute average improvement of 2.16% accuracy over the state-

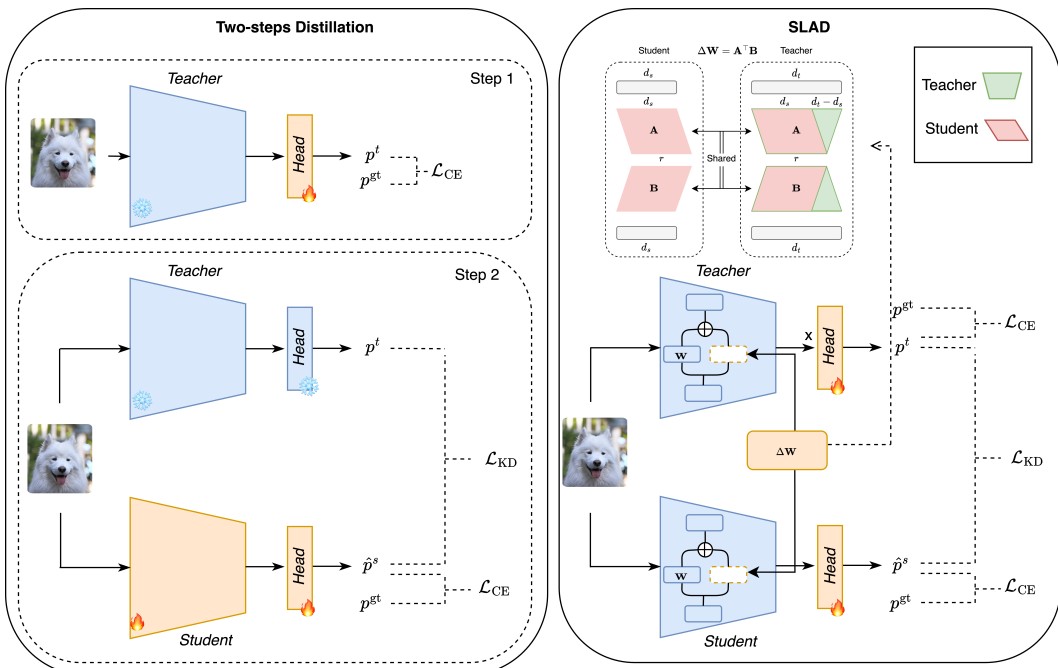

Figure 2: **Overview of our SLAD method**. (left) The two-step distillation introduced in Marrie et al. (2024) where the teacher is first probed on the downstream task and then distilled into a student. (right) The SLAD method where the teacher and the student are trained jointly in one step while sharing their LoRA adapter weights.

of-the-art classification benchmarks while being 2 times faster than the full-fine-tuning strategy (Figure 1).

**Summary of our contributions:**

- We investigate why a fine-tuning of the teacher during the two-step task-specific distillation process results in worse knowledge distillation. Our analysis shows key insights on harmful feature misalignment between the teacher and the student that occurs during fine-tuning of the teacher.

- Leveraging this insight, we propose to use low-rank adaptation techniques to achieve a better trade-off between capturing task-specific intrinsics and knowledge distillation.

- We propose the SLAD method, which shares a subset of adapters of the teacher with the student while jointly training them. Through a wide range of experiments, SLAD achieves better knowledge distillation to the student while being significantly faster than the previous state-of-the-art on multiple classification and segmentation benchmarks.

## 2 RELATED WORK

This section first examines previous research on task-agnostic and task-specific distillation, then explores advancements in parameter-efficient fine-tuning (PEFT).

### 2.1 GENERIC KNOWLEDGE DISTILLATION

Knowledge Distillation (KD) is a widely adopted compression approach (Wang & Yoon, 2022; Gou et al., 2021) that transfers knowledge from a large "teacher" model to a smaller "student" model, allowing the student to achieve similar performance with significantly reduced complexity. The foundational work by Ba & Caruana (2014); Hinton et al. (2015) introduced soft-label distillation, where

the student learns from both hard labels and softened teacher outputs. Building on this, numerous KD variations have emerged. FitNets (Romero et al., 2015), for example, incorporated intermediate supervision to transfer knowledge from hidden layers. Other methods have introduced distinct approaches: attention map distillation (Zagoruyko & Komodakis, 2017), relational distillation between data points (Park et al., 2019), distribution-based distillation (Huang & Wang, 2017), and intermediate "teacher assistants" to bridge the teacher-student gap (Mirzadeh et al., 2019). However, these techniques often target task-specific models. With the recent rise of task-agnostic self-supervised learning (SSL) models, which can be adapted to various downstream tasks through simple methods like linear probing, there is a growing need for KD approaches that effectively leverages the knowledge of bigger SSL models while adapting the smaller models to downstream tasks. This distinction has led to the development of task-specific distillation methods, which differ from classical KD by adapting both teacher and student to the same downstream task rather than compressing a fixed, fully-trained teacher.

## 2.2 Task Specific Distillation

Task-specific distillation has emerged as a key approach to tailor knowledge distillation for agnostic foundation models trained with self-supervised learning (SSL). TinyBERT (Jiao et al., 2020) introduced this approach in NLP, followed by G2SD (Huang et al., 2024) in computer vision, both employing a two-step distillation process. First, a general distillation is applied to yield a smaller, foundation model from the teacher. Then, the teacher undergoes fine-tuning on a downstream task, followed by task-specific distillation. More recently, Marrie et al. (2024) demonstrated that probing the teacher directly often yields better results than fine-tuning. Building on this, our method integrates LoRA adapters, enhancing model flexibility while preserving agnostic knowledge from pretraining. We further show that sharing these adapters between teacher and student models promotes better knowledge transfer, improving performance across both models.

## 2.3 Parameter Efficient Fine-Tuning

To address the substantial costs of fine-tuning large models, particularly foundation models, parameter-efficient fine-tuning (PEFT) (Lialin et al., 2024; Han et al., 2024) techniques have gained popularity by updating only a subset of the model's weights (or dimensions) while keeping others frozen. Visual Prompt Tuning (VPT)(Jia et al., 2022) introduces additional tokens to the input patches and optimizes only these new tokens, while BitFit(Ben Zaken et al., 2022) restricts fine-tuning to bias terms. LayerNorm tuning (Kim et al., 2022; De Min et al., 2023) proceeds by fine-tuning only the scale and bias parameters of the LayerNorm modules of the model. Adapt-Former (Chen et al., 2022) takes a similar approach, adding lightweight adapter layers trained exclusively for task-specific adjustments. Singular Value Fine-tuning(SVF) (Sun et al., 2022) starts by applying SVD decomposition to the model's weights and then updates only the singular values of the weight matrices while keeping the rest frozen.

Among PEFT approaches, Low-Rank Adaptation (LoRA)(Hu et al., 2022) stands out as a prominent method that inserts low-rank adapters parallel to the model weights, training only these adapters. LoRA has demonstrated strong performance across various domains and tasks. Recent studies (Kopiczko et al., 2024) have explored layer-wise sharing of LoRA adapters. However, to the best of our knowledge, we are the first to investigate sharing LoRA adapters weights across different models, particularly within a knowledge distillation framework.

## 3 Method

In this section, we first remind the reader with the two-step distillation proposed by Marrie et al. (2024). We then introduce our first contribution using LoRA for task-specific distillation before exposing the details of our proposed method SLAD which enhances knowledge transfer between the teacher and student models.

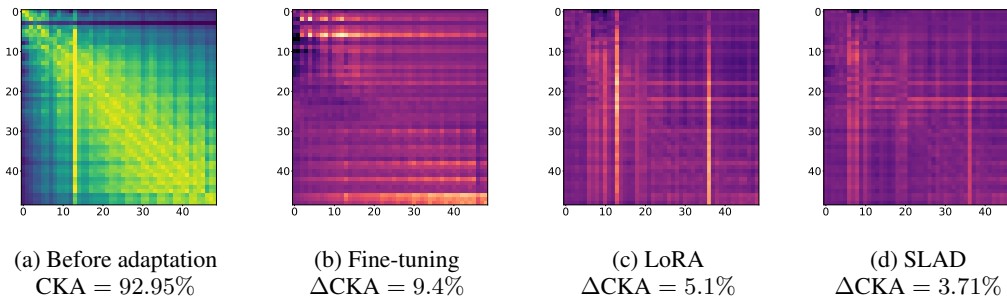

(a) Before adaptation
CKA = 92.95%

(b) Fine-tuning
$\Delta$CKA = 9.4%

(c) LoRA
$\Delta$CKA = 5.1%

(d) SLAD
$\Delta$CKA = 3.71%

Figure 3: Centered Kernel Alignment between representations of a teacher ViT-B and a student ViT-S on CUB dataset. We plot CKA similarities between all pairs of layers of the two encoders. (a) CKA of pre-trained backbones (before adaptation on the downstream task). (b-c-d) Difference of CKA before and after adaptations for different adaptation techniques. The initial mean CKA value across similar layers is 92.95%. When fully fine-tuning the teacher, the CKA value drops by 9.4%, on the other hand using LoRA only deteriorates the CKA by 5.09%. When sharing the weights of the LoRA adapters between the teacher and the student model, our method SLAD achieves a CKA of 89.65%, with a drop of 3.71% indicating better feature alignment during the distillation process, which benefits the knowledge distillation transfer.

### 3.1 TWO-STEP DISTILLATION

We consider a downstream task $\mathcal{D}_{\text{train}} = \{\mathbf{x}_i, \mathbf{y}_i\}_{i=1}^N$, where $\mathbf{x}$ are training images and $\mathbf{y}$ are the associated labels (either for classification or segmentation). Let $e_t$ and $e_s$ be two pre-trained (agnostic) teacher and student encoders respectively. Task specific distillation as introduced in Marrie et al. (2024) folds in two steps. First, a teacher model $f_t$ is trained, where $f_t = p_t \circ e_t$ and $p_t$ is an adaptation function, typically a linear head. The teacher is trained using the empirical risk minimization loss:

$$\mathcal{L}_{task}(f_t) = \frac{1}{N} \sum_{i=1}^N \ell_{\text{task}}(f_t(\mathbf{x}_i), \mathbf{y}_i), \tag{1}$$

where $\ell_{\text{task}}$ is typically the Cross-Entropy loss.

The second step consists of training the student model $f_s = p_s \circ e_s$ using $\mathcal{L}_{\text{task}}(f_s)$ alongside a divergence loss between the student's predictions and the teacher's:

$$\mathcal{L}_{\text{KD}}(f_s) = \alpha_s \underbrace{\mathcal{L}_{\text{task}}(f_s)}_{T=1} + \alpha_{\text{KL}} T^2 \underbrace{D_{\text{KL}}(f_s, f_t)}_{T>1}, \tag{2}$$

where $T$ represents the softmax temperature, $D_{\text{KL}}$ is the Kullback-Leibler (KL) divergence (Kullback & Leibler, 1951) and $(\alpha_s, \alpha_{\text{KL}})$ are weighting hyperparameters.

Here, one can envision multiple strategies to train the student and the teacher. For instance, during the adaptation of the teacher, the encoder $e_t$ can either be frozen, commonly known as linear probing (Alain & Bengio, 2017), or entirely fine-tuned. Experimental results in Marrie et al. (2024) showed that while fine-tuning the teacher's encoder yields better performance for the teacher on the downstream tasks, this results in a less efficient knowledge distillation for the student compared to freezing the encoder and learning the projection $p_t$ alone. This means that although fine-tuning a teacher allows to capture more information on the downstream task, it does not translate to a better teacher for the student.

We argue that fine-tuning all the weights of the teacher in the first step leads to a mismatch in feature representations between the teacher and student encoders, resulting into a lower knowledge distillation compared to a frozen and task-agnostic teacher. This is commonly observed when fine-tuning a model on a downstream task, often leading to catastrophic forgetting (Kirkpatrick et al., 2017), modifying the intrinsic feature representations of the model. Our analysis in Figure 3 using Centered Kernel Alignment (CKA) (Kornblith et al., 2019) shows that fine-tuning all weights of the

teacher results in worse alignment of the feature representations between the two encoders, when compared to other approaches including LoRA. This mismatch arguably hinders the knowledge distillation process.

Although linear probing (Alain & Bengio, 2017) the teacher, as suggested in Marrie et al. (2024), achieves better knowledge distillation by preserving feature alignment between the two encoders, it does not capture the task-relevant knowledge compared to a fully fine-tuned model, resulting in a suboptimal knowledge transfer. In the following, we propose to design a solution that balances this trade-off.

### 3.2 Task-Specific Distillation With Low Rank Adaptation (LoRA)

A recent approach to balance catastrophic forgetting and performance is through the use of low-rank adaptation (Wistuba et al., 2024; Liang & Li, 2024). We thus propose to adapt first the teacher on the downstream task using LoRA adapters, and then distill knowledge on the student using LoRA adapters again.

For a weight matrix $\mathbf{W}_0 \in \mathbb{R}^{m \times n}$, LoRA introduces low-rank adapters $\mathbf{A} \in \mathbb{R}^{m \times r}$ and $\mathbf{B} \in \mathbb{R}^{r \times n}$:

$$\mathbf{W} = \mathbf{W}_0 + \Delta\mathbf{W} = \mathbf{W}_0 + \mathbf{AB} \qquad (3)$$

The teacher model is first trained on the task using LoRA, updating only the classifier weights and LoRA adapters. Subsequently, task-specific knowledge is distilled into the student model (also equipped with LoRA adapters and a classifier, with other parameters frozen) using Equation 2.

This strategy allows the use of better adapters than linear probing without heavily disrupting the feature representation between the teacher and the student, as shown in Figure 3, hence allowing better knowledge distillation and better adaptation of the student model.

While the use of LoRA adapters allows smaller degree of freedom for disruptive feature alignment between the two encoders, the distillation process is limited to the transfer of logits and therefore it does not necessarily enforce the feature alignment. In the following, we propose another strategy to further enforce feature alignment between the teacher and the student.

### 3.3 SLAD : Shared LoRA Weights

One possible strategy to enforce the feature alignment during distillation is to use an additional consistency loss between the features of the teacher and the student as performed in Jiao et al. (2020) for natural language processing. However, the teacher and student are typically of different sizes often resulting in different dimensionality of the features, which hinders a straightforward design of a consistency loss and would require learning an additional projection layer during the distillation process.

Instead, we build on the previously proposed LoRA adaptation technique. Given that the adaptation weights are initialized from scratch, we share these weights between the two encoders resulting in our final proposed method SLAD as depicted in Figure 2. Differences in the internal dimensions between the teacher and student encoders necessitate an adaptive approach to sharing: indeed, the LoRA adapters in the student model are smaller than those of the teacher. Therefore, we share only a subset of the teacher's LoRA weights (Figure 4), matching the dimensions of the student adapters.

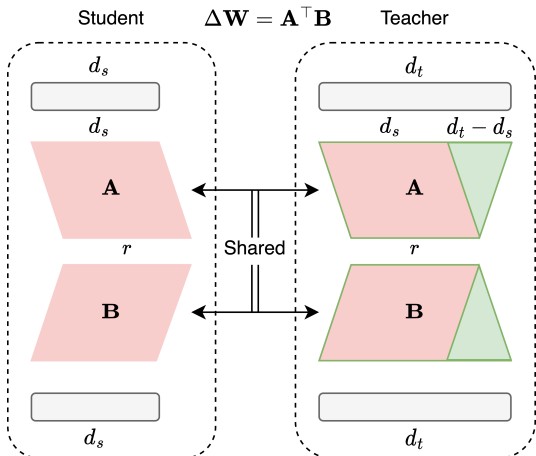

Figure 4: Overview of the sharing mechanism for SLAD : the teacher features with dimension $d_t$ goes through LoRA adapters $A$ and $B$ while the student features with dimension $d_s \leq d_t$ goes through a subset of the LoRA adapters corresponding to the first $d_s$ dimensions.

Given that the adaptation weights are shared between the teacher and the encoder, we propose to simultaneously train the two models on the task. Each model is optimized on the target using its respective classifier, while the teacher transfers knowledge to the student through a KL divergence loss. The overall loss function is defined as:

$$\mathcal{L}_{\text{SLAD}} = \alpha_s \underbrace{\mathcal{L}_{\text{task}}(f_s)}_{T=1} + \alpha_t \underbrace{\mathcal{L}_{\text{task}}(f_t)}_{T=1} + \alpha_{\text{KL}} T^2 \underbrace{D_{\text{KL}}(f_s, f_t)}_{T>1}, \tag{4}$$

where $(\alpha_s, \alpha_t, \alpha_{\text{KL}})$ are weighting hyperparameters.

This strategy enables the teacher to develop robust task-specific representations while transferring this knowledge to the student adapters through both knowledge distillation and weight sharing. As a result, the student benefits from better feature alignment as depicted in Figure 3, resulting in better task-relevant representations and enhancing the effectiveness of the distillation process.

Furthermore, this joint training not only refines the student's ability to capture nuanced representations but also alleviates the need for a two-step distillation process enabling simultaneous teacher-student training and allowing for direct, one-step adaptation, making it both faster and more effective. We demonstrate the effectiveness of this approach in the next section.

## 4 EXPERIMENTS

### 4.1 DATASETS

We evaluated our method on the same types of tasks as Marrie et al. (2024), including fine-grained classification, and semantic segmentation. For fine-grained classification, we conducted experiments on CUB (Wah et al., 2011) with 200 bird species, FGVC Aircraft (Maji et al., 2013) with 100 aircraft classes, and DTD (Cimpoi et al., 2014) with 47 texture classes. For semantic segmentation, we tested on Cityscapes (Cordts et al., 2016; 2015) with 19 classes.

### 4.2 IMPLEMENTATION DETAILS

Following the setup of Marrie et al. (2024). We provide implementation details in this section and refer the reader to the appendix for additional details such as data augmentation strategies and prediction head design.

#### 4.2.1 BACKBONE MODELS

For our models, we utilize DINOv2 (Oquab et al., 2023), with all models being Vision Transformers (ViT) (Dosovitskiy et al., 2021). Specifically, we use ViT-L with 300M parameters and ViT-B with 86M parameters as teacher models, and ViT-B and ViT-S with 21M parameters as student models. In our experimental setup, we therefore can assume that both the teacher and student are already trained in a task-agnostic manner. Our focus, as outlined above, is on adapting the student backbone model to downstream tasks.

#### 4.2.2 IMPLEMENTATION DETAILS

**Training Procedure.** We followed the training protocol outlined in Marrie et al. (2024). However, due to the unavailability of their source code, we re-implemented their methodology. All models were trained for 30 epochs, except in the case of distillation from a probed teacher, where training was extended to 80 epochs, consistent with Marrie et al. (2024). We employed the AdamW optimizer with a cosine learning rate schedule. Hyperparameters such as learning rate and weight decay were selected via grid search based on validation performance. In scenarios where a validation set was not provided, we followed Marrie et al. (2024) and reserved 10% of the training data for validation.

For the Cityscapes dataset, where test labels are not publicly available, we report results on the validation set and use 10% of the training data as a validation split. For knowledge distillation, we used a temperature of $T = 2$ and loss weights $(\alpha_s, \alpha_{\text{KL}}) = (0.5, 0.5)$. In LoRA-based training, we

set the rank to $r = 16$, initializing the matrix $A$ using Kaiming uniform initialization and the matrix $B$ with zeros. LoRA was applied to the attention matrices $QKV$.

In our shared training framework, SLAD, we used a uniform mapping function with equal weighting $(\alpha_s, \alpha_t, \alpha_{KL}) = (1, 1, 1)$. We explore the effect of different mapping functions in the ablation study presented in Section 5.2.1. All experiments were conducted using three fixed random seeds, and we report averaged results. For evaluation, we report classification accuracy (acc) for fine-grained classification tasks and mean Intersection over Union (mIoU) for segmentation tasks.

# 5 RESULTS

## 5.1 MAIN RESULTS

In Table 4, we report the performance across various classification and segmentation tasks. Distilling using LoRA consistently outperforms linear probing. This result shows that training the teacher using low-rank adaptation (LoRA) allows to achieve better knowledge distillation than previous state-of-the-art results (Marrie et al., 2024).

Table 1: **Distillation/SLAD results of different teachers and different students of DINOv2** for fine-grained classification and semantic segmentation. We report accuracy for classification and mIoU for segmentation. Probing* refers to our re-implementation of the method proposed in Marrie et al. (2024).

| Student | Teacher | Method | Fine-grained classification (acc) | | | | Semantic segmentation (mIoU) |
|---|---|---|---|---|---|---|---|
| | | | CUB | Aircraft | DTD | Avg. | Cityscapes |
| ViT-S | ViT-B | Probing* | 89.80 | 89.44 | 81.93 | 87.06 | 69.32 |
| | | LoRA (ours) | 90.01 | 91.82 | **84.71** | 88.85 | 72.75 |
| | | SLAD (ours) | **90.54** | **92.50** | 84.56 | **89.20** | **73.91** |
| ViT-S | ViT-L | Probing | 89.70 | 89.20 | 83.40 | 87.43 | 74.00 |
| | | Probing* | 89.61 | 90.10 | 81.61 | 87.11 | 72.42 |
| | | LoRA (ours) | 90.04 | 91.46 | 84.18 | 88.56 | 73.43 |
| | | SLAD (ours) | **90.44** | **92.44** | **84.24** | **89.04** | **73.97** |
| ViT-B | ViT-L | Probing* | 90.61 | 91.02 | 83.49 | 88.37 | 73.87 |
| | | LoRA (ours) | 91.41 | 93.25 | **86.58** | 90.41 | 77.06 |
| | | SLAD (ours) | **91.81** | **94.18** | 86.40 | **90.80** | **77.35** |

Furthermore, our results highlight the effectiveness of the weight-sharing strategy, SLAD achieves state-of-the-art performance with an average increase of 2.71% accuracy on fine-grained classification tasks for the ViT-L to ViT-S setting. Furthermore, SLAD achieves state-of-the-art performance on Cityscapes, a semantic segmentation task compared to our re-implementation of probing. Additionaly, thanks to its joint training strategy, SLAD boosts existing benchmarks while being faster to train, as depicted in Figure 1. These results are consistent with our feature alignment analysis in Figure 3, where SLAD exhibits higher representational similarity between the teacher and student compared to both LoRA and fine-tuning. This supports our claim that improved feature alignment contributes to more effective knowledge distillation.

## 5.2 ABLATION STUDIES

### 5.2.1 CHOICE OF MAPPING FUNCTION

LoRA weight sharing between teacher and student models is performed at the layer (block) level. However, as the teacher and student may differ in depth (e.g., ViT-L has 24 blocks, while ViT-B and ViT-S have 12), a mapping function is required to align student blocks to teacher blocks. We consider three strategies: (1) **First**, where each student block is matched to the corresponding block in the first $n$ teacher blocks (with $n$ being the number of student blocks); (2) **Last**, where student blocks are matched to the last $n$ teacher blocks; and (3) **Even**, where student blocks are mapped to uniformly spaced teacher blocks. In our setting (ViT-L to ViT-B/S), the even mapping corresponds to the function $g(i) = 2i$, where $i$ is the student block index and $g$ is the mapping function.

Table 2: Accuracies of different mappings functions on the CUB, Aircraft, and DTD datasets.

| Student | Teacher | Mapping | Fine-grained classification (acc) | | |
|---|---|---|---|---|---|
| | | | CUB | Aircraft | DTD |
| ViT-S | ViT-L | *Even* | **90.44** | 92.44 | 84.24 |
| | | *First* | 90.32 | **92.64** | 84.29 |
| | | *Last* | 90.33 | 92.58 | **84.38** |
| ViT-B | ViT-L | *Even* | 91.81 | **94.18** | 86.40 |
| | | *First* | 91.81 | 93.95 | **86.67** |
| | | *Last* | **91.82** | 94.04 | **86.67** |

Table 3: **Teacher Performance in Fine-Grained Classification and Semantic Segmentation.** Comparison between ViT-{B,L} models trained directly on downstream tasks and their performance when utilized as teacher models in SLAD.

| Model | Method | Fine-grained classification (acc) | | | Semantic segmentation (mIoU) |
|---|---|---|---|---|---|
| | | CUB | Aircraft | DTD | Cityscapes |
| ViT-B | Probing | 90.29 | 87.35 | 83.76 | 69.54 |
| | LoRA | 90.64 | 92.86 | 86.28 | 77.90 |
| | SLAD | 91.50 | 93.50 | 86.17 | 75.45 |
| ViT-L | Probing | 91.46 | 90.84 | 84.40 | 72.58 |
| | LoRA | 91.78 | 94.32 | 87.23 | 79.99 |
| | SLAD(ViT-S) | 92.21 | 94.96 | 87.21 | 77.61 |
| | SLAD(ViT-B) | 92.57 | 95.14 | 87.52 | 79.02 |

Table 2 presents the performance of each block mapping strategy. We observe that the **Even** mapping yields the best results on CUB, while performing slightly worse on DTD. This variation suggests that different tasks may benefit from information located at different depths of the teacher model, consistent with findings in Jiao et al. (2020). Nonetheless, the overall performance gap between the three mappings remains small, indicating that model performance is relatively insensitive to the specific choice of mapping function.

### 5.3 TEACHER PERFORMANCE

As shown in Table 3, the teacher trained with SLAD outperforms training with LoRA alone, with gains up to 0.8 % accuracy. This indicates that the teacher benefits from joint training with the student, even though the latter is a distilled version. This observation is consistent with Deep Mutual Learning (DML) (Zhang et al., 2018; Gou et al., 2024), where models collaboratively learn and improve together. We also find that SLAD with a stronger student (e.g., ViT-B) yields larger gains for the teacher than with a weaker one (e.g., ViT-S), suggesting that student capacity directly influences the teacher's benefit.

## 6 CONCLUSION

In this paper, we propose to investigate the existing two-step task-specific distillation process. Previous results showed that a probed teacher yields better distillation results on the student than a fully fine-tuned one. Our findings shows that balancing the expressivity of a teacher and its knowledge distillation can be achieved through better feature alignment between the teacher and the student. We propose different methods to enhance the alignment, showing significant improvements for task-specific distillation. Notably, leveraging low-rank adapters via parameter-sharing between the two encoders can further improve the state-of-the-art of task-specific distillation for both classification and semantic segmentation.

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

## A   APPENDIX

## B   COMPARISON WITH FINE-TUNING

In this section, we report the comparison of our method with fine-tuning. Table 4 shows that our method SLAD outperforms fine-tuning across different classification datasets. This highlights the effectiveness of our method to perform knowledge distillation better than linear probing and full fine-tuning strategies.

Table 4: **Distillation/SLAD results of ViT-L teacher and ViT-S student of DINOv2** for fine-grained classification. We report accuracy for each datasets in fine-grained classification and their average accuracy. Probing* refers to our re-implementation of the method proposed in Marrie et al. (2024).

| Student | Teacher | Method | CUB | Aircraft | DTD | Average |
|---------|---------|--------|-----|----------|-----|---------|
| ViT-S | ViT-L | Fine-tuning (Marrie et al., 2024) | 88.60 | 88.90 | 81.50 | 86.33 |
| | | Probing (Marrie et al., 2024) | 89.70 | 89.20 | 83.40 | 87.43 |
| | | Probing* ($Marrie\ et\ al.$, 2024) | 89.61 | 90.10 | 81.61 | 87.11 |
| | | LoRA (ours) | 90.04 | 91.46 | 84.18 | 88.56 |
| | | SLAD (ours) | **90.44** | **92.44** | **84.24** | **89.04** |

## C   ADDITIONAL IMPLEMENTATION DETAILS

### C.1   PREDICTION HEAD

Following Marrie et al. (2024), we employ a two-layer MLP as the prediction head for classification tasks and a linear head for segmentation, consisting of batch normalization followed by a convolutional layer with a kernel size of 1. For fine-grained classification, we concatenate the CLS token from the last three blocks, while for segmentation, we use the patch embedding from the final block. Let $n_{in}$, $n_{hidden}$, $n_{out}$ respectively denote the number of input, hidden and output neurons in the MLP, Following Marrie et al. (2024), we set $n_{hidden} = n_{in}$ for ViT-S and $n_{hidden} = \sqrt{n_{in} \times n_{out}}$ for ViT-B and ViT-L.

## C.2 DATA AUGMENTATION

For fine-grained classification, we applied the following augmentations during training: `Resize`, `CenterCrop`, `ColorJitter`, and `RandomHorizontalFlip`. For validation and testing, we resized and center-cropped all images to $512 \times 512$ from $640 \times 640$. For the Cityscapes dataset, training augmentations included `RandomResize` to $640 \times 640$, `RandomCrop` to $560 \times 560$, `RandomHorizontalFlip`, and `RandomPhotometricDistortion`. At validation and test time, we applied a sliding window approach with crops of $560 \times 560$ and a stride of $s = 280$. Following the recommendation of Beyer et al. (2022), we ensured that both teacher and student models received the same batch of images with identical data augmentations applied.

## D LIMITATIONS

Based on our analysis, balancing expressivity of the teacher and knowledge distillation can be achieved through better feature alignment. While LoRA adapters enhances feature alignment compared to fine-tuning and expressivity compared to probing, its low-rank nature has been showed to be less flexible than a fully fine-tuned teacher (Albert et al., 2025). In future work, we will explore how to enhance both expressivity and knowledge distillation of the teacher model, ensuring better knowledge transfer. Additionally, our insights on feature alignment offer multiple options for the choice of the shared layers as well as potential regularization losses to enhance the alignment allowing better knowledge transfer.

## E TRAINING TIME

Beyond performance improvements, SLAD provides notable advantages in training efficiency. Conventional distillation typically follows a two-stage process: training the teacher for $n$ epochs, followed by distillation into the student over an additional $m$ epochs. Resulting in a total training budget of roughly $n + m$ epochs.

SLAD, by contrast, performs joint optimization of the teacher and student within a single training phase of $n$ epochs. Although each epoch is computationally heavier, due to forward and backward passes through both models, the total time is significantly reduced.

For instance, on the Aircraft dataset, distillation from a ViT-B to a ViT-S using the standard pipeline takes approximately 8.3 hours (2.8 hours for teacher training and 5.5 hours for distillation over 30 epochs) on a A100 GPU. In contrast, SLAD completes joint training in just 5.6 hours, underscoring the substantial efficiency gains achieved by SLAD. Moreover, Marrie et al. (2024) report using 80 epochs for distillation, which increases the total training time to 17.5 hours.

## F RUNTIME DETAILS.

The majority of our experiments were conducted on A100 GPUs with 40GB of RAM, while the remaining runs used V100 GPUs. The runtime varied significantly depending on the task and model size, for example, Probing with ViT-S on the CUB dataset took approximately 2 hours, whereas SLAD training between ViT-L and ViT-B on CUB required up to 12 hours.

