# OpenReview forum: "SLAD : Shared LoRA Adapters for Task Specific Distillation"
_ICLR.cc/2026/Conference — ICLR 2026 Conference Withdrawn Submission_

### Official Review · Reviewer_xbnQ · 2025-10-23

**Soundness:** 2
**Presentation:** 2
**Contribution:** 1
**Rating:** 2
**Confidence:** 4

**Summary:**

This paper targets *task-specific distillation*: given a large (teacher) and a small (student) version of the same pretrained backbone, both must be adapted to a downstream task while transferring the teacher’s task-specific knowledge to the student. The authors first observe that full fine-tuning of the teacher improves its own accuracy yet reduces representation alignment with the student, degrading distillation; conversely, linear probing preserves alignment but limits the teacher’s expressiveness. To trade off these extremes, they propose (1) adapting both models with LoRA and (2) jointly training them while sharing a subset of LoRA weights, explicitly encouraging feature alignment (the method is termed SLAD). The overall objective combines individual task losses for teacher and student with a temperature-scaled KL-distillation loss (Eq. 4). Experiments on CUB, FGVC-Aircraft, DTD, and Cityscapes show improvements over pure probing or standalone LoRA-based distillation, with reported training-time savings versus a two-stage pipeline.

**Strengths:**

- **Simple yet practical**: By sharing a subset of LoRA adapter weights between teacher and student (layer-to-layer mapping + dimension-wise truncation), SLAD retains the efficiency of PEFT while explicitly encouraging feature alignment—easy to implement in production.
- **Clear motivation and evidence for alignment**: Figure 3 quantifies alignment with CKA: full fine-tuning drops ΔCKA ≈ 9.4 %, LoRA alleviates it to ≈ 5.1 %, and SLAD pushes it further to ≈ 3.7 %. The progressive alignment gain mirrors the accuracy improvement, yielding a coherent causal story.
- **Multi-task validation**: Evaluated on three fine-grained classification datasets plus semantic segmentation, SLAD almost uniformly outperforms probing and vanilla LoRA distillation; e.g., ViT-L → ViT-S shows consistent gains on classification and a student boost on Cityscapes.
- **Thorough ablation on layer mapping**: Compares First / Last / Even alignment strategies between student and teacher layers; results are similar, with Even slightly ahead, providing reassurance that the method is not sensitive to the exact mapping choice.

**Weaknesses:**

- **Lack of Novelty.** The paper fine-tunes with LoRA and performs knowledge-distillation via parameter sharing. However, fine-tuning a small-scale dataset with LoRA is an intuitive engineering practice rather than a methodologically novel contribution. Consequently, labelling vanilla LoRA as “(ours)” in Figure 1 and Table 1 is inappropriate; please remove the “(ours)” tag or explicitly clarify which components are truly introduced by this work.
- **Additional experimental:** At present, LoRA is only injected into the QKV weight matrices of the attention layers. To fully validate the generality of the proposed approach, the authors are encouraged to run supplementary experiments that additionally apply LoRA to other parameter blocks—e.g., the FFN layers (such as fc1/fc2 or gate/up/down projections)—and report the corresponding comparisons.
- **Lack of ablation experiments:** Equation (4) introduces three loss coefficients—αₛ, αₜ, and α_KL—yet the experiments adopt the uniform setting (all ones) without any ablation. The authors should systematically vary these weights to demonstrate how each term influences final performance and justify the chosen balance.
- **Inconclusive empirical claims:** Section 5.3 argues that joint training with SLAD “improves the teacher,” but Table 3 shows the opposite trend on half of the datasets. For ViT-B, SLAD outperforms vanilla LoRA on CUB and Aircraft yet underperforms—often by a large margin—on DTD and Cityscapes; the average accuracy is therefore lower than with standard LoRA. The same pattern is observed for ViT-L when ViT-S serves as the student. These results do **not** support the authors’ conclusion that SLAD consistently boosts teacher performance; at best, the effect is dataset-dependent. Please either revise the claim or provide additional evidence that isolates when and why the teacher benefits.
- **Training-speedup claim is unsupported.** The paper advertises “≈2× acceleration” over standard fine-tuning, yet Appendix E offers no controlled time breakdown. It is unclear whether the comparison was run under identical hardware, batch-size, and implementation. The 30-epoch figure for the baseline is ambiguous: does it refer to the first stage only or to the sum of both teacher and student stages? The epoch budget for the joint SLAD training is not reported at all. The raw numbers (8.3 h vs. 5.6 h) amount to only **1.48×** speed-up, far below the claimed 2×.
- **Writing issue**: The paper spends considerable space arguing that full-parameter fine-tuning harms distillation performance. This observation has already been well-documented in prior work, which likewise proposes remedies such as linear probing. Please condense or remove these reiterations and reallocate space to clearly emphasize the **novel** focus of the present study.

**Questions:**

See weaknesses.

---

### Official Review · Reviewer_r7y2 · 2025-10-24

**Soundness:** 2
**Presentation:** 2
**Contribution:** 2
**Rating:** 2
**Confidence:** 5

**Summary:**

This paper investigates task-specific knowledge distillation, starting from the observation that fine-tuning a teacher model can harm knowledge transfer to a student due to feature representation mis-alignment. The authors propose SLAD, a method that uses Low-Rank Adaptation (LoRA) and shares LoRA adapter weights between the teacher and student during a single-stage joint training process. This is intended to enforce better feature alignment, leading to better student performance and faster training compared to traditional two-stage methods.

**Strengths:**

1. The proposed method is easy to implement.
2. The paper is clear written.

**Weaknesses:**

1. The paper's primary motivation rests on the claim that "mis-alignment in feature representation" causes a fine-tuned teacher to be a worse knowledge distiller, but there does exist not a direct causality. This hypothesis requires more validation to rule out other confounding factors.
2. The authors invoke the concept of catastrophic forgetting from continual learning to describe the representational shift during fine-tuning. However, the setting in this paper is standard transfer learning, not continual learning.
3. The discussion on general knowledge distillation (Section 2.1) is outdated.
4. The experimental evaluation is insufficient in several key areas: 1) Missing comparison to general KD approaches; 2) Missing comparison to other PEFT methods; and 3) Lack of hyperparameter analysis.

**Questions:**

refer to weakness

---

### Official Review · Reviewer_UhU5 · 2025-11-01

**Soundness:** 3
**Presentation:** 3
**Contribution:** 2
**Rating:** 6
**Confidence:** 3

**Summary:**

This paper introduces SLAD (Shared LoRA Adapters for Task-Specific Distillation), a method designed to improve knowledge distillation from large to small vision models in downstream tasks. The authors identify a key issue in existing distillation pipelines: fine-tuning the teacher model improves its performance but disrupts feature alignment with the student, reducing distillation effectiveness. In contrast, linear probing preserves alignment but limits the teacher's ability to capture task-specific features.

**Strengths:**

1. Simple yet effective remedy: Combining LoRA with weight-sharing yields a single-stage distillation procedure that is both faster and more accurate; no extra modules or heavy hyper-parameter tuning are required.

2. Empirical thoroughness: Experiments span three fine-grained classification sets and one segmentation benchmark, multiple teacher/student pairs (ViT-L/B/S), and ablations on adapter mapping, showing consistent gains over probing, full fine-tuning and vanilla LoRA.

3. Win-win property: Joint training not only boosts the student but also lifts the teacher, a result rarely reported in distillation literature.

**Weaknesses:**

1. Small-scale downstream tasks: Evaluation is restricted to three medium-sized classification sets (≤ 12 k images) and one segmentation set; no large-scale or multi-task results are offered.

2. Low-rank capacity ceiling: LoRA’s expressiveness is intentionally limited; the paper concedes that a fully fine-tuned teacher can still outperform LoRA teachers in isolation, hinting that SLAD may sacrifice peak teacher accuracy for alignment.

3. In fact, it's necessary to clarify the benefits of sharing parameters between the student and teacher classes, and the difference between using two lora instances (one dedicated, one shared), similar to: https://openreview.net/forum?id=G1Hlubz1fR

**Questions:**

1. What are the concrete benefits of sharing LoRA parameters between teacher and student (as in SLAD) versus simply giving each model its own LoRA adapter, and how would performance differ if we adopted a hybrid scheme—e.g., one dedicated LoRA instance per model plus an additional shared LoRA block—as proposed in works like https://openreview.net/forum?id=G1Hlubz1fR?

2. Does the enforced alignment that comes from sharing low-rank adapters force the teacher to sacrifice the higher accuracy it could reach with full fine-tuning, and how much peak performance is lost in exchange for better distillation?

3. How well would SLAD’s shared-LoRA distillation hold up on large-scale or multi-task benchmarks (when num of tasks > 100), given that all reported tests are on relatively small datasets (≤12 k images each)?

---

### Note · Authors · 2025-11-13

I have read and agree with the venue's withdrawal policy on behalf of myself and my co-authors.